# Design of a Broadband Perfect Solar Absorber Based on a Four-Layer Structure with a Cross-Shaped Resonator and Triangular Array

Yushan Chen [1], Kewei You [1], Jianze Lin [1], Junwei Zhao [2], Wenzhuang Ma [1], Dan Meng [1], Yuyao Cheng [1] and Jing Liu [1,*]

[1] School of Ocean Information Engineering, Jimei University, Xiamen 361021, China
[2] School of Electronic Engineering, Jiangsu Ocean University, Lianyungang 222005, China
* Correspondence: jingliu@jmu.edu.cn

**Abstract:** As solar energy is a low-cost and clean energy source, there has been a great deal of interest in how to harvest it. To absorb solar energy efficiently, we designed a broadband metamaterial absorber based on the principle of Fabry–Pérot (FP) cavities and surface plasmon resonance (SPR). We propose a broadband perfect absorber consisting of a four-layer structure of silica–tungsten–silica–titanium ($SiO_2$–W–$SiO_2$–Ti) for the incident light wavelength range of 300–2500 nm. The structure achieves perfect absorption of incident light in the wavelength range of 351.8–2465.0 nm (absorption > 90%), with an average absorption of 96.3%. The advantage of our proposed structure is that it combines the characteristics of both high and broadband absorption, and has high overall absorption efficiency for solar radiation. It is also independent of polarization and insensitive to incident angle. We investigated how absorption was affected by different structures, materials, geometric parameters, and refractive indices for different dielectric materials, and we explored the reasons for high absorption. This structure is refractory and ultrathin, and it offers a good tradeoff between bandwidth and absorption. It therefore has premium application prospects and value.

**Keywords:** perfect absorber; ultra-broadband; visible region; near-infrared region

## 1. Introduction

As the global demand for energy has increased dramatically, burning of fossil fuels has led to many environmental problems, such as the greenhouse effect. To minimize the serious consequences of global warming and keep the planet's average surface temperature from rising by more than 2 °C, the current direction of sustainable development is toward significantly increasing the share of renewable energy in the global energy structure and doubling energy efficiency [1]. Solar energy is a premium and cheap renewable energy source. Many studies have been carried out on the acquisition, storage, and use of solar energy [2–6]—especially how to collect it. There is a simple way to obtain high broadband absorption: use blackbody paint. However, premium infrared calibration blackbody sources have remained relatively costly. Commercial blackbody paint is an option, but it can only absorb part of sunlight [7].

Based on a survey of the last decade's literature, we found that metamaterials have many extraordinary properties, because the materials are human-selected and their parameters are artificially designed. Metamaterials thus offer a wide range of application scenarios for achieving various purposes through different designs—for example, thermal energy emission [8,9], detection [10–12], sensing [13,14], and biological and biomedical applications [15,16]. Other than that, metamaterials could also play an important role in the aerospace field, because of their small size, robustness against vibrations, and high degree of integration [17,18]. In addition, metamaterial absorbers can be very effective for solar harvesting [19,20].

Because noble metals can provide strong plasmonic resonance and optical coupling, many studies have used them as solar absorbers [21–23]. However, the intrinsic absorption of noble metals means that the overall bandwidth of such absorbers is narrow. Li et al. proposed an 11-layer absorber whose optical properties could be tailored by manually changing the parameters of a periodic Au array, but its absorption bandwidth was relatively narrow (400–700 nm) [21]. Qin et al., also using gold, designed an absorber with three absorption peaks to achieve broadband absorption in the 1000–2500 nm band. However, the absorber had insufficient absorption in the visible-light range, and the average absorption rate was less than 70% [22]. Ali Elrashidi et al. designed a three-layer absorber composed of Ti, $SiO_2$, and Ag after trying a variety of materials. This structure achieved a broadband absorption of 1410 nm [23].

Because the intrinsic absorption of noble metals makes it difficult to achieve broadband absorption, and because of their scarcity, high cost, and poor chemical and physical stability, researchers are paying less attention to precious metals and turning to other types of materials, such as refractory metals. Liu et al. designed a titanium nitride (TiN) nanoresonator array and coated it on a titania ($TiO_2$) array; together, these provided multiple resonant modes for the entire structure. After a series of parameter adjustments, an absorption bandwidth of 1110 nm was obtained [24]. Yu et al. designed a solar absorber with high thermal stability using a refractory metal. The intrinsic absorption performance and plasmonic resonance of titanium provided 1264 nm broadband absorption [25].

Using silicon nitride as a dielectric material, Liu et al. piled four different-sized layers of iron—from large to small—obtaining four absorption peaks, and achieving 1.4–11.4 μm absorption (>60%). Although the bandwidth of this structure was large, the average absorption rate was low [26]. Wu et al. used Ti and $SiO_2$ materials with good thermal stability and achieved broadband absorption from 0.596 to 4.102 μm. However, the structure had poor absorption performance for incident light in the visible region—especially in the 500–750 nm wavelength range [27]. The aforementioned absorbers had either high absorption or broadband absorption, but not both. We believe that the most effective absorption of sunlight can only be achieved if both broadband and high absorption can be achieved. In designing our absorber, we avoided noble metals and chose tungsten and titanium—two metals with excellent overall performance. Table 1 shows the difference between the proposed structure and other structures.

**Table 1.** Comparisons between perfect absorbers.

| Researchers | Bandwidth (nm) | Structure | Reference |
|---|---|---|---|
| Li et al. | 300 nm | 11-layer with periodic Au array | [17] |
| Qin et al. | 1500 nm | made up of a Au nano-cuboids array | [18] |
| Ali Elrashidi and colleagues | 1410 nm | composed of Ti, $SiO_2$, and Ag array | [19] |
| Liu et al. | 1110 nm | TiN nano-resonator array under $TiO_2$ array | [20] |
| Yu et al. | 1264 nm | array with an elliptical TiN plate and a $TiO_2$ elliptical plate of the same size | [21] |
| Liu and other researchers | 10 μm | piled four different sized layers of iron, from large to small | [22] |
| Wu et al. | 3.506 μm | consists of silicon dioxide colloidal crystal array and Ti | [23] |
| us | 2113.2 nm | four-layer structure of silica-tungsten-silica-titanium ($SiO_2$-W-$SiO_2$-Ti) | this work |

We propose an ultra-wideband perfect absorber for solar light based on the finite-difference time-domain (FDTD) method. The FDTD method is based on Maxwell equations, widely used for modeling nanophotonic devices, processes, and materials. The basic idea of the FDTD method is to divide space into meshes and obtain the continuous field distribution by solving discrete numerical solutions on the mesh grids. Previous researchers

have demonstrated that actual experimental results are largely identical to those obtained using the FDTD method [28,29]. Therefore, it was feasible to probe the absorption efficiency of our proposed structure with this method. The whole structure is centrosymmetric and axisymmetric, making it independent of polarization as well as insensitive to the incident angle. The attenuation properties of tungsten and titanium mean that they have a high imaginary refractive index. Combining the two metals achieved high absorption and broadband absorption at the same time. Both materials also have high melting points, contributing to the structure's overall thermal stability. We can use electron beams to deposit the substrate of this absorber, and employ lithography and different etching processes to build cross-shaped resonators and triangular arrays. The structure can be prepared on a large scale due to the absence of noble metal materials. In general, our proposed absorber has the advantages of simple structure, easy preparation, good absorption performance, and thinness, so it holds promise for applications.

## 2. Model and Parameter Design of the Perfect Solar Absorber

We took full advantage of the absorber's high absorption principle to improve the absorption efficiency and bandwidth. The cross structure of tungsten and the four surrounding triangular tungsten cylinders easily provided powerful surface plasmon resonance. The two layers below ($SiO_2$ and Ti) cooperated with tungsten to form the FP cavity, which is generally composed of two layers of metals and a dielectric layer between the metals. This can constrain the incident light and lead to strong interaction between the incident light and the whole structure, further enhancing the absorption of incident light in the visible and near-infrared bands. In addition, we attached a layer of $SiO_2$ to the tungsten array to enhance the plasmon resonance, augmenting the absorption of incident light. The refractive index values for tungsten and titanium were obtained from the *CRC Handbook of Chemistry and Physics* [30], and the refractive index of $SiO_2$ was set to 1.45 [31]. We used FDTD Solutions to verify our proposed structure. Periodic conditions were set in the X and Y directions, and PML boundary conditions were set in the Z direction. The optimized material parameters were l = 410 nm, w = 90 nm, t = 150 nm, P = 700 nm, $h_1$ = 80 nm, $h_2$ = 200 nm, $h_3$ = 60 nm, and $h_4$ = 200 nm. A schematic diagram of the structure, along with a top view and side view, is shown in Figure 1.

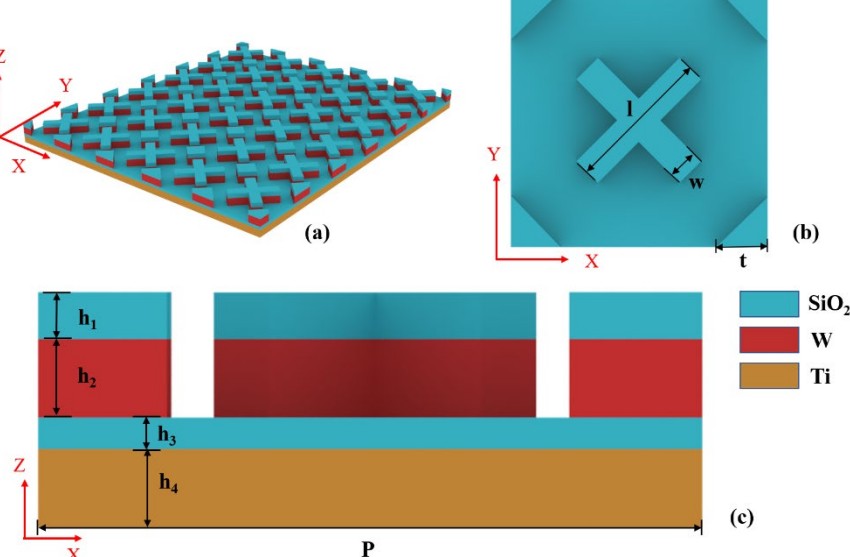

**Figure 1.** (**a**) Schematic diagram of the periodic structure. (**b**) Top view of the absorber with its parameters, where l = 410 nm, w = 90 nm, and t = 150 nm. (**c**) Side view of the absorber with its parameters, where P = 700 nm, $h_1$ = 80 nm, $h_3$ = 60 nm, and $h_2$ = $h_4$ = 200 nm.

## 3. Results and Discussion

The formula used to calculate the absorption of the absorbing structure was $A = 1 - R - T$, where $R$ is the normal reflectance. We set the thickness of the titanium substrate to 200 nm, which resulted in almost no electromagnetic waves penetrating the entire structure, so, $T = 0$; therefore, $A = 1 - R$. We thus calculated the absorption of the structure for incident light in the wavelength range of 300–2500 nm. As shown in Figure 2a, the absorption curve of this structure possessed three absorption peaks, located at 800, 1320, and 2120 nm, respectively. The highest absorption value was located at the incident light wavelength of 1320 nm, with a maximum absorption of 99.4%. The perfect absorption region of this structure (absorption > 90%) covered almost the entire visible–near-infrared band, with a perfect absorption bandwidth of 2113.2 nm (from 351.8 to 2465.0 nm). Within this perfect absorption band, we calculated the average absorption using the following formula, where $\lambda_2$ = 351.8 nm and $\lambda_1$ = 2465.0 nm:

$$A = \int_{\lambda_2}^{\lambda_1} A(\lambda) d\lambda / (\lambda_1 - \lambda_2) \tag{1}$$

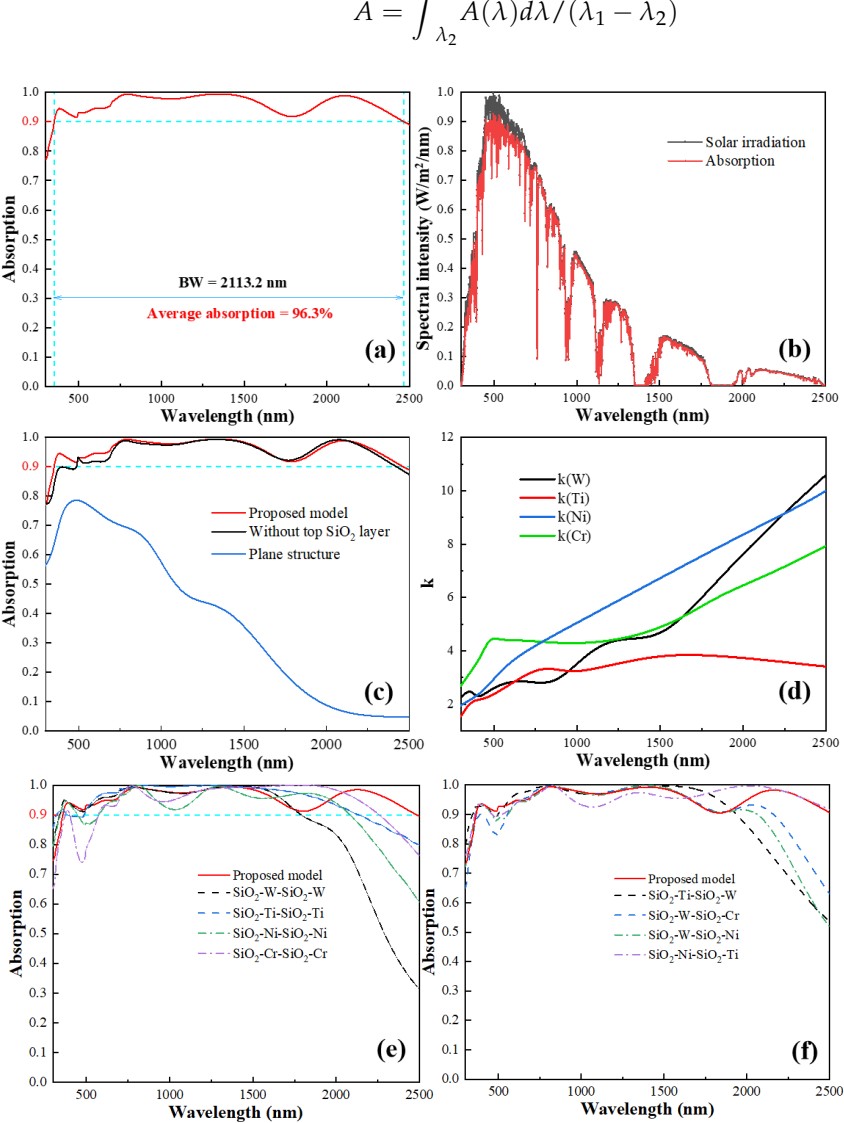

**Figure 2.** (**a**) Absorption curve of the proposed model. (**b**) Solar spectral intensity and absorption curve. (**c**) Absorption of the proposed model, model without the top SiO$_2$ layer, and planar structure, which is continuous on the *x*-axis and *y*-axis. (**d**) Imaginary refractive index of W, Ti, Ni, and Cr. (**e**,**f**) Absorption of the proposed model and models using other combinations of metallic materials.

We used the calculated average absorption to determine the energy absorption of this structure for solar scattering. The results are shown in Figure 2b. The proposed structure had only a small energy loss in the visible and ultraviolet (UV) bands, and absorbed almost all of the sunlight in the other bands, indicating its good absorption performance.

We also explored how the absorptivity was affected by a planar structure, the absence of a top dielectric structure, and the use of different materials. The structure with a $SiO_2$ top layer had a smoother absorption curve than one without, meaning that the $SiO_2$ top layer brought better solar coupling to the whole structure. As shown in Figure 2c, the structure with a $SiO_2$ top layer had higher absorption efficiency in the visible wavelength band. The presence of a $SiO_2$ dielectric made it easier for electromagnetic waves with smaller wavelength values to be absorbed by the whole structure. Within the solar radiation spectrum, the visible band accounts for most of the solar energy, so high absorption efficiency in the visible band must be ensured. Planar structures with similar dimensions achieved only about 80% absorption when the wavelength of incident light was close to 500 nm. Comparison of the absorption curves for the proposed model and the planar structure showed that the presence of an artificially designed cross-shaped resonator and a triangular array had a significant impact on the absorption bandwidth and peak. This is also supported by previous experimental results—absorbers with special shapes (e.g., discs) are more efficient than absorbers with stacked structures [26,32].

Figure 2e,f show that the use of different materials caused the whole absorption curve to go up and down at different wavelengths of incident light. The entire curve also underwent redshift or blueshift, depending on the materials. Choosing metal materials with a high imaginary refractive index is helpful to provide high-quality (high-Q) resonances [33], and it is evident that although W, Ti, Ni, and Cr all have a high imaginary refractive index (see Figure 2d), using tungsten and titanium for the top array and substrate, respectively, optimized the structure's absorption.

This was because the cross-shaped resonator, triangular array, and substrate led to incomplete overlap of the absorption bands. Using only one metal material resulted in a narrower overall absorption bandwidth and a lower overall average absorption rate. Choosing different materials caused the entire absorption peak to appear in different regions, so selecting other materials for the top array as well as the substrate maximized the absorption and yielded the widest bandwidth. From Figure 2f, we can see that the absorption of the structure using other metal combinations is not as good as that proposed here, because its absorption bandwidth and average absorption are lower, or the absorption rate in the visible-light region is lower. By selectively changing different materials and conducting simulation experiments, we believe that the proposed model ($SiO_2$–W–$SiO_2$–Ti) can provide the most perfect absorption.

SPRs (surface plasmon resonances) consist of two types of resonance—localized surface plasmon resonance (LSPR), and propagating surface plasmon resonance (PSPR)—and the intensity of SPRs is due to many factors, such as the size of the structure, the refractive index of the chosen material, etc. [33]. The reason that our proposed structure has broadband high absorption is that it can produce high-Q resonances in the incident light range of 300–2500 nm.

To explain why this absorbing structure has broadband and efficient absorption, we scanned the electric and magnetic fields using FDTD Solutions when the incident light was vertically incident on this structure without polarization at specific wavelengths; the results are presented in Figure 3a–i. Among these, we selected the specific wavelengths 800, 1320, and 2120 nm, corresponding to the three peaks in the absorption curve. When the wavelength of incident light is 800, 1320, or 2120 nm, the damping of resonance is the largest, which is helpful for us to observe the distribution of electric and magnetic fields.

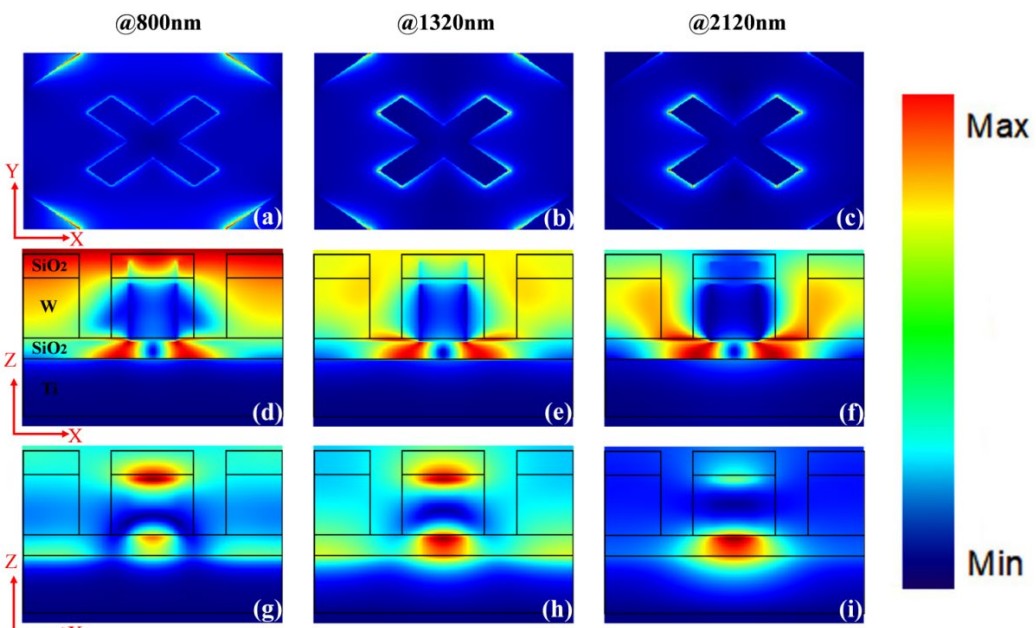

**Figure 3.** (**a**–**c**) Electric field scanning in the X–Y plane. (**d**–**f**) Electric field scanning in the X–Z plane. (**g**–**i**) Magnetic field scanning in the X–Z plane.

By scanning the electric field in the X–Y plane, as shown in Figure 3a–c, we found that as the incident light's wavelength increased, the electric field intensity at the boundary of the triangular array decreased, while the electric field intensity at the edge of the cross-shaped resonator increased. This was because the electric field was generated by LSPR under subwavelength conditions. LSPR is a phenomenon whereby electrons and photons interact with one another in metals under subwavelength conditions, and eventually produce high-intensity electric and magnetic fields. The dimensions of the cross-shaped resonator were slightly larger than those of the triangular array, and both were smaller than the incident light's wavelength. Therefore, as the incident light's wavelength increased, the LSPR gradually increased around the central cross-shaped resonator, and gradually decreased around the triangular array. Thus, our proposed structure can interact with the incident light in a large wavelength range, producing a very strong LSPR.

By scanning the electric field in the X–Z plane, as shown in Figure 3d–f, we found that a strong electric field was also excited in the $SiO_2$ dielectric layer below the cross-shaped resonator. A strong LSPR was generated above silicon dioxide, around tungsten, resulting in the generation of strong electric field.

Under subwavelength conditions, the continuous adjacent lossless dielectric layer and the metal produced strong resonance with the incident light. This resonance is called PSPR, which produces strong electric and magnetic fields. The FP cavity caused the interaction between the incident and reflected light from the metal surface in the dielectric layer, and finally generated a strong electric and magnetic field. Thus, the generation of the electric field shown in Figure 3d–f was the result of the joint action of LSPR, PSPR, and the FP cavity.

By observing the magnetic field distribution in the X–Z plane, as shown in Figure 3g–i, we also assessed the combination of PSPR and the FP cavity. Similarly, the magnetic field in the $SiO_2$ dielectric layer was generated from cooperation between the PSPR and the FP cavity. Since the cross-shaped resonator and the triangular array were covered by lossless dielectric layers, a strong PSPR was generated. With the increase in the wavelength of the incident light, the proposed structure could always produce a strong magnetic field, although the magnetic field changed slightly.

In Figure 3d–i, we can see that strong electric and magnetic fields are excited. The causes of the excitation of the electric and magnetic fields are similar—both include the combination of PSPR and the FP cavity—but the strong electric field and magnetic field

excited by the FP cavity only exist between Ti and W, while the strong electric field and magnetic field excited by PSPR exist around the $SiO_2$ layer, above the W array.

In our design, adding lossless dielectric layers above and below the cross-shaped resonator and the triangular array effectively provided high-Q SPRs [33]. Moreover, our chosen material (tungsten) has a large imaginary refractive index and possesses strong intrinsic metal loss, causing the whole structure to provide strong SPRs. In order to reach high and broadband absorption, we should try to achieve high-Q plasmonic resonances.

In Figure 4, we explore what the cross-shaped resonator and the triangular array contributed to the absorption for the same parameters. In the incident light wavelength range of 300–500 nm, the absorption curves of the proposed model, the structure without a cross-shaped resonator, and the structure without a triangular array had similar shapes, and the average absorption of the proposed model was the highest. The three absorption curves peaked successively at an incident wavelength of about 800 nm. We believe that the combined effect of the triangular array and the cross-shaped resonator located the absorption peak of the complete structure in the middle of the two, and the absorption of the complete structure was provided by the cross-shaped resonator and the triangular array together. At an incident light wavelength of about 2000 nm, the absorption curve of the structure without a triangular array peaked, while the absorption curve of the structure without a cross-shaped resonator slowly decreased. The combined effect of the two resulted in an absorption peak when the wavelength of incident light was 2100 nm. Although this absorption peak was slightly lower than the peak produced by the structure without the triangular array, it was broad, resulting in a bandwidth of perfect absorption greater than 2000 nm. We conclude that combining a cross-shaped resonator and a triangular array can facilitate the effects of plasmon resonance and the FP cavity, resulting in better absorption of sunlight.

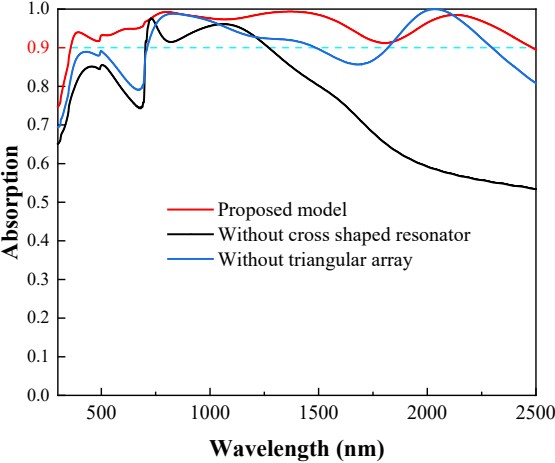

**Figure 4.** Comparison of the absorption curves of the structure without a cross-shaped resonator, the structure without a triangular array, and the proposed model.

After analyzing the reasons for the structure's good absorption, we know that the sizes of the cross-shaped resonator and triangular array are closely related to its performance, because the size of the structure does have a certain influence on its resonance [34], and SPR only exists when the scale of the structure is far less than the wavelength of incident light. [33]. To obtain the best microwave absorption performance, we performed a series of sweeps of the cross-shaped resonator and triangular array.

Figure 5a shows the absorption performance of the resonator for different length and width conditions. As the resonator's size increased, there was a significant redshift of the absorption peak; when the length (l) and width (w) increased from 350 and 50 nm to 470 and 130 nm, respectively, the wavelength value at the absorption peak increased from approximately 1850 to 2500 nm. Conversely, as the size decreased, the absorption was

higher when the wavelength of the incident light was smaller. Since energy scattered by the sun is mainly concentrated in sunlight, whose wavelength ranges from 300 to 2500 nm, to achieve both high and broadband absorption, we chose a compromise size: l = 410 nm and w = 90 nm. When we observed the effects on absorption caused by changing the triangular array parameters (Figure 5b), similar results were seen—the position of the absorption peak gradually redshifted when the side length (t) increased from 110 to 190 nm. We chose t = 150 nm as the final parameter for working with the cross-shaped resonator to achieve the best solar absorption performance.

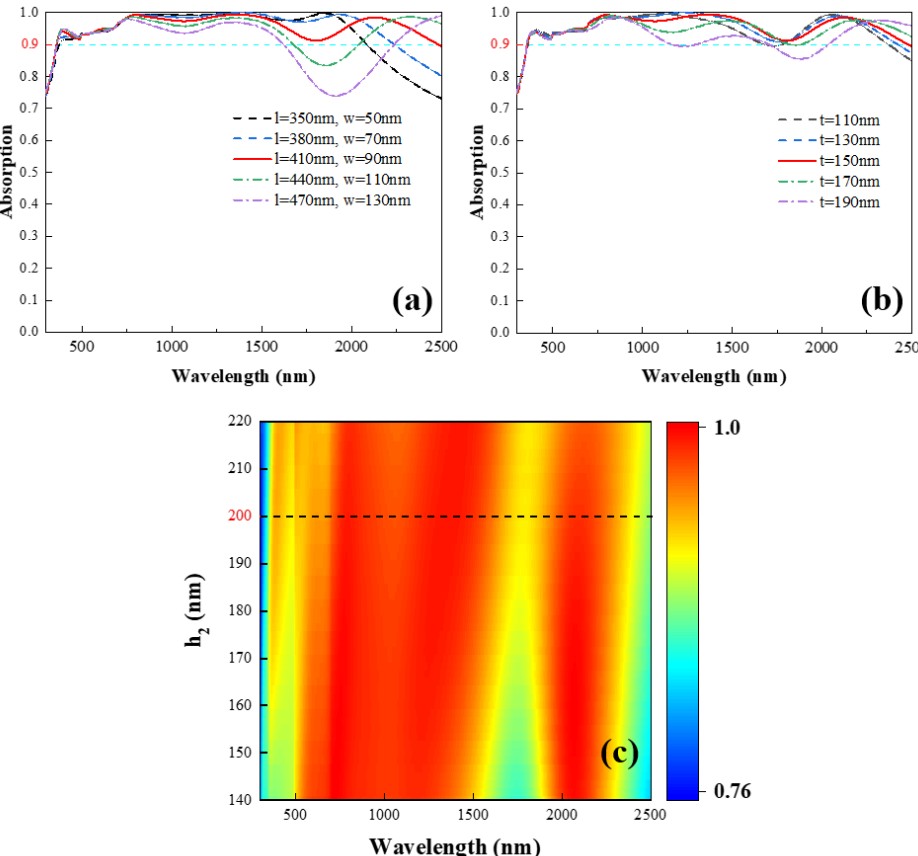

**Figure 5.** (**a**) Absorption of cross-shaped resonators with different lengths (l) and widths (w). (**b**) Absorption of triangular arrays with different side lengths (t). (**c**) Cross-shaped resonator and triangular array height scan ($h_2$).

Figure 5c shows that as the height of the cross-shaped resonator and the triangular array increased, the width of the two absorption peaks gradually increased, but when the height of the tungsten layer increased to a certain level, the width of the absorption peak gradually stopped expanding, and the peak of the absorption peak decreased slightly. To achieve the best absorption effect, we chose 200 nm as the height of both, which yielded an absorption bandwidth above 2000 nm and good absorption in the visible and UV regions (300–760 nm).

To explore which material (magnesium fluoride ($MgF_2$), $SiO_2$, alumina ($Al_2O_3$), silicon nitride ($Si_3N_4$), or $TiO_2$) was the most suitable lossless dielectric for this structure, we compared their absorption properties, as shown in Figure 6a. The materials' refractive indices were taken from the *Handbook of Optical Constants of Solids* [31]. We found that $SiO_2$—a very common material—gave the best solar absorption performance for the whole structure. We therefore chose to use it as the lossless dielectric material for this structure.

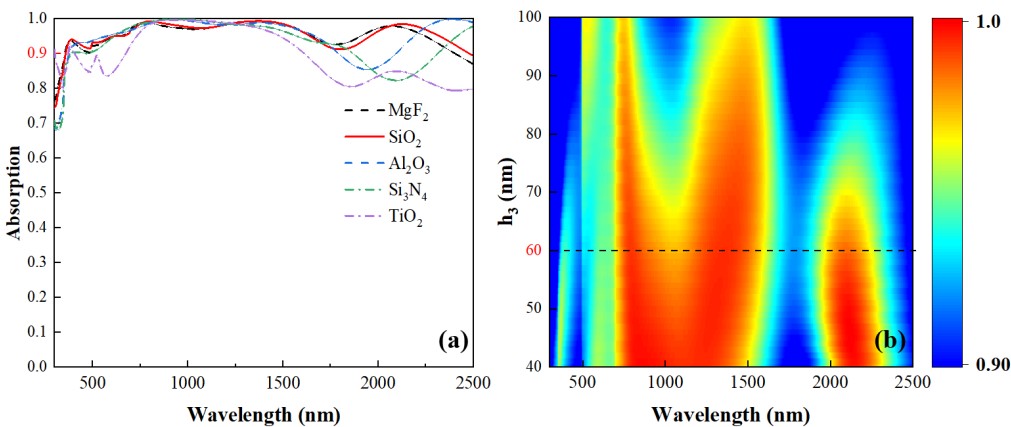

**Figure 6.** (**a**) Absorption of different dielectric materials: polyethylene, silica, quartz, vanadium dioxide, and alumina. (**b**) Thickness sweep of the lossless dielectric layer.

Based on previous electric and magnetic field scans, we knew that in order to have good solar absorption performance, we needed to excite the strongest plasmon resonance from the lossless dielectric layer under the cross-shaped resonator and the triangular array. A proper dielectric thickness ($h_3$) delicately combined the joint action of the FP cavity and PSPR to achieve the most effective absorption. We swept the height of $h_3$ and highlighted the regions where the absorption was above 90%. In Figure 6b, the more the absorption exceeds 90%, the closer the color is to red, while the regions where the absorption is below 90% are uniformly blue to highlight the parameters that optimized the absorption. As $h_3$ increased, the width of the high absorption region first gradually increased, and then decreased. We knew that there were two wide absorption regions and a narrow absorption peak. As the thickness of the lossless dielectric layer increased, the two broad absorption regions tended to gradually approach one another, and then gradually moved away after the closest distance. As the radiation intensity in the incident light wavelength range of 300–500 nm was relatively high, we chose $h_3$ = 60 nm as the thickness of the dielectric layer to achieve perfect absorption.

To theoretically assess why our proposed structure has good solar absorption performance, we analyzed the structure using effective medium theory. This theory can be used to calculate whether the impedance of the proposed structure meets the impedance-matching principle and whether the proposed structure has good absorption efficiency in theory. This research method has been widely used in previous work [35]. The following formulae can be used to calculate the structure's impedance [36,37]:

$$Z = \pm \sqrt{\frac{(1+S_{11})^2 - S_{21}{}^2}{(1-S_{11})^2 - S_{21}{}^2}} \tag{2}$$

$$S_{11} = S_{22} = \frac{i}{2}\left(\frac{1}{Z} - Z\right)\sin(nkd) \tag{3}$$

$$S_{21} = S_{12} = \frac{1}{\cos(nkd) - \frac{i}{2}\left(Z + \frac{1}{Z}\right)\sin(nkd)} \tag{4}$$

where $S_{11}$, $S_{22}$, $S_{21}$, and $S_{12}$ are all S parameters, where $S_{11}$ = R and $S_{12}$ = T = 0; k is the wave vector, and d is the thickness of the designed structure. In addition, an inhomogeneous structure can be replaced conceptually by a continuous material, and there would be no difference in the scattering characteristics between the two [36]. The continuous material is characterized by effective refractive index n (used in Equations (3) and (4)), which can be calculated by $n = ((\varepsilon_m \times \varepsilon_d)/(\varepsilon_m + \varepsilon_d))^{1/2}$, where $\varepsilon_m$ is the permittivity of the metal materials, and $\varepsilon_d$ is the permittivity of the nearby dielectrics [38]. In free space, when

the plane wave is transmitted to the surface of the medium, the medium's reflectivity is as follows:

$$R = \frac{Z_0 - Z_{in}}{Z_0 + Z_{in}} \tag{5}$$

where $Z_0$ is the impedance of the free space, and $Z_{in}$ is the impedance of the incident material in contact with the free space. In general, the real part of the impedance in free space is 1, and the imaginary part is 0. According to the impedance-matching principle, when the impedance of the material is close to that of the free space, the reflection coefficient is very small. Since $T$ and $R$ are very small and $A = 1 - R - T$, when $A$ is close to 1, excellent absorption performance can be obtained for the whole structure. In Figure 7, we can see that in the region from visible to near-infrared, the real part of the structure's impedance is close to 1, and the imaginary part is close to 0. Therefore, we believe that this structure has good absorption efficiency in free space.

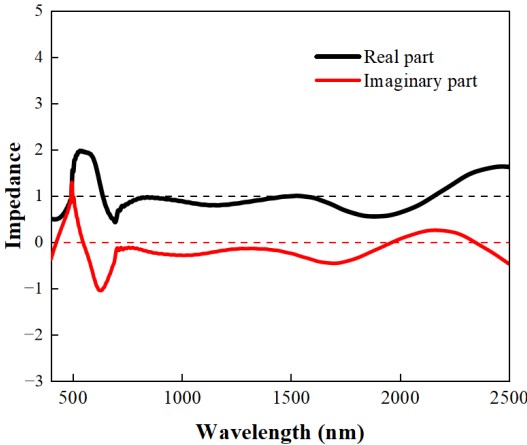

**Figure 7.** The impedance of the proposed structure.

When this absorber is actually put into use, the light is not always vertically incident without polarization. We therefore scanned the structure at different polarization angles and different incident angles. The results in Figure 8a show that there was little change in the structure's absorption with increasing polarization angle during the change from TE to TM polarization (0° to 90°). This was because our proposed structure is not only axisymmetric, but also centrosymmetric. The high degree of symmetry kept the overall absorption stable under different conditions of polarized light incidence. Moreover, the absorber maintained perfect absorption in the incident light wavelength range of 700–2050 nm, and high absorption (>80%) in the incident light wavelength range of 300–2350 nm, with different and increasing incidence angles. We thus believe that this structure would be able to absorb solar energy efficiently in practical application scenarios.

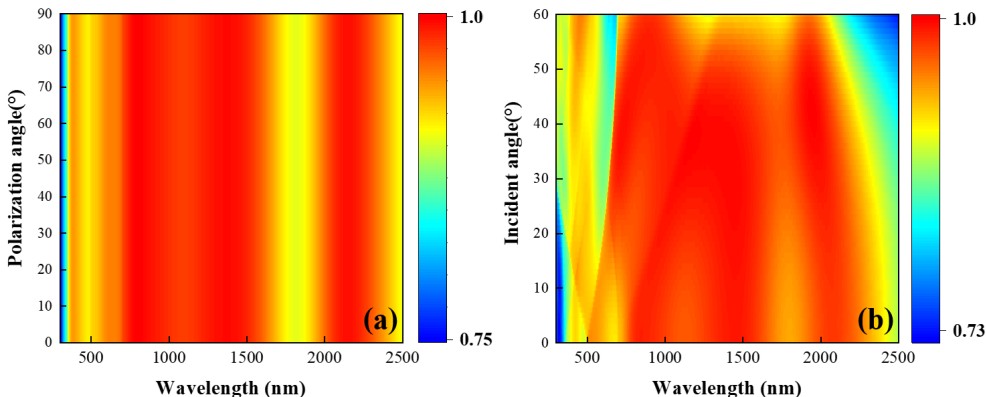

**Figure 8.** (**a**) Absorptivity at different incident angles. (**b**) Absorptivity at different polarization angles.

## 4. Conclusions

In this paper, we propose a solar absorber based on the FDTD method that can absorb solar energy across the visible and near-infrared regions. The width of the perfect absorption band reaches 2113.2 nm, within which the average absorption is 96.3% and the maximum is 99.4%. Through theoretical analysis and electromagnetic field scanning, we explain why the structure is able to achieve broadband absorption, and compare the effects of different materials and dimensions on the overall absorption. Finally, sweep diagrams of the polarization angle and incident angle show that the structure is independent of the polarization angle and insensitive to the incident angle, allowing it to be used in practical application scenarios. With the increasing demand for renewable energy, this absorber could play a major role in solar energy absorption, thermoelectric conversion, and many other applications.

**Author Contributions:** Conceptualization, Y.C. (Yushan Chen), and K.Y.; methodology, K.Y.; software, J.L. (Jianze Lin); validation, W.M., D.M. and Y.C. (Yuyao Cheng); formal analysis, J.L. (Jing Liu); investigation, D.M.; resources, J.L. (Jing Liu); data curation, Y.C. (Yuyao Cheng); writing—original draft preparation, K.Y.; writing—review and editing, D.M.; visualization, J.Z. and W.M.; supervision, Y.C. (Yushan Chen); project administration, J.L. (Jing Liu); funding acquisition, Y.C. (Yushan Chen). All authors have read and agreed to the published version of the manuscript.

**Funding:** This work was supported by the Fujian Provincial Natural Science Foundation 2020J01712; the Second Youth Talent Support Program of Fujian Province (Eyas Plan of Fujian Province 2021); the Fujian Provincial Department of Science and Technology 2019H0022; the Science Fund for Distinguished Young Scholars of Fujian Province 2020J06025; the Youth Talent Support Program of Jimei University ZR2019002; the Innovation Fund for Young Scientists of Xiamen under Grant 3502Z20206021; and the Xiamen Marine and Fishery Development Special Fund 20CZB014HJ03.

**Institutional Review Board Statement:** Not applicable.

**Informed Consent Statement:** Not applicable.

**Data Availability Statement:** No data were generated or analyzed in the present research.

**Conflicts of Interest:** The authors declare that there are no conflicts of interest related to this article.

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
