# Peer review of "Design of a Broadband Perfect Solar Absorber Based on a Four-Layer Structure with a Cross-Shaped Resonator and Triangular Array"

_photonics, doi:10.3390/photonics9080565_

Round 1

Reviewer 1 Report

The result of broadband nearly perfect absorption at almost full solar spectrum has large impact.

Previous studies are properly referred to describe the advantage of this study.

Therefore, I think the manuscript could be published if proper revision is made.

The points should be considered are shown below.

 -          The authors states that one of the reasons for broadband absorption is the use of different metals in the top and bottom layers. However, the explanation is not clear. The reason why broadband absorption can be obtained by using different metals should be clearly shown in terms of the optical constants of each metal.

In this study, the combination of tungsten and titanium is shown, but is the same phenomenon observed for tungsten and nickel, titanium and nickel, or any other combination

-          The authors states that another reason for broadband absorption is the existence of multiple resonance modes. The characteristic resonance modes at three representative wavelengths are shown by electromagnetic field analysis. However, I think it is still insufficient as an explanation for broadband absorption. Even at wavelengths in between each resonant mode, high absorption rates are observed Why? Is it because there are other resonance modes or because the damping of each resonance mode is large? Please add clear explanations.

-          The authors mention that “this was the result of the joint action of the propagating surface plasmon resonance (PSPR) and the FP cavity” at L166 and also “The magnetic field in the SiO2 dielectric layer was generated from cooperation between the PSPR and the FP cavity” at L169. However, how we can understand the coupling of these resonances from Fig. 3? The detail explanations and/or additional analysis should be added to describe the coupling of these resonances.

Also, according to the explanation of each resonance, height of tungsten h2 shouldn’t have any effect on the absorption characteristics. However, the authors mention “the height of the cross-shaped resonator and triangular array increased, the width of the two absorption peaks gradually increased, but when the height of the tungsten layer increased to a certain level, the width of the absorption peak gradually stopped expanding, and the peak of the absorption peak decreased slightly.” Why? Please add clear explanations.

-          The authors make discussion using effective medium theory to describe broadband absorption. However, this is very strange. If effective medium theory can explain this, then the resonance effect is irrelevant. In other words, if the effective refractive index is similar, a broad absorption bandwidth can be obtained from any structure, and it means the resonance effect is irrelevant. It is necessary to clarify which effect contributes to broadband absorption.

-          Comparisons have been made with other metamaterials in the introduction. However, I think the simplest way to obtain broadband high absorption is to use blackbody paint. Is there any advantage over this? If so, it should be properly explained.

Followings are minor corrections.

-          Is analyzed reflectance total reflectance or normal reflectance? It should be mentioned.

-          In Fig. 2, the description “Plane structure” is unclear. It should be explained more in the caption. Also, the description “Original model” is bit strange. “Proposed model” might be better.

-     In the title, “struc-ture” should be corrected

Reviewer 2 Report

The paper fits to the line of papers describing efforts addressed to obtaining perfect absorbers that can be used in solar harvesting. These absorbers are based on metamaterials, in this case the metamaterial is composed of four layers of periodically ordered crosses and triangles of tens-nanometer size and spaces beetwen them are treated as Fabry-Perot interferometers. These assumptions are taken into account in the numerical simulation calculating the absorption spectrum via the FDTD method. The results are very promising, yielding absorption spectra that reach an average absorbance more than 96% in the spectral range of about 2000 nm. The paper is rather well written, and conclusions are clear. However I have a couple of comments of technical character.

Firstly, when the FDTD method is first time mentioned it would be fine to add some two sentences on this method, simply what is it about. It will be very useful for reader who does not have to be exactly on the subject.

Secondly, when the Authors describe scanning the reader does not know whether it is done experimentally or only in the simulation, it should be clearly said. It is especially important since the Authors say e.g. on “observing the magnetic field distribution” in this context.

And finally I can only express my hope that sooner or later such a perfectly absorbed structure will be realized in practice.

Summarising, I think that the presented paper can be published in the Photonics, after taking my comments into account.

Reviewer 3 Report

The Authors propose a design of broadband perfect solar absorber, based on four layers with cross-shaped resonator and triangular array. The results have been carried out by using FDTD approach. The proposed device is very interesting and the manuscript deserves the publication after addressing the following comments:

-          In the Introduction section, the Authors should highlight the benefits of metasurfaces, proposed for imaging (see, as example, Metasurfaces for biomedical applications: imaging and sensing from a nanophotonics perspective. Nanophotonics10(1), 259-293, 2021), bio (see, as example, Exploring the limit of multiplexed near-field optical trapping. Acs Photonics8(7), 2060-2066, 2021), detection (see, as example, Metasurface enabled quantum edge detection. Science advances6(51), eabc4385, 2020.). This approach could increase the potential appeal of the proposed manuscript, as example of metasurface application.  

-          A crucial aspect regards also the application field. A solar absorber could represents a disruptive technology also in the Space field for the green migration of satellites. However, the radiation hardness should be taken into account (see, Measured radiation effects on InGaAsP/InP ring resonators for space applications. Optics Express27(17), 24434-24444, 2019; Radiation hardness of high-Q silicon nitride microresonators for space compatible integrated optics. Optics express22(25), 30786-30794.2014). Please comment on it.

-          A comparison of the proposed device with others reported in literature should be provided. A table could be useful.

Minor comments:

-          All figures should be resized.

-          The quality of Fig. 1 should be improved.

Round 2

Reviewer 1 Report

The author has properly addressed almost all of my comments.

Therefore, I think the manuscript can be published. 

One thing that I would like to make sure is how the value of n used in equations (3) and (4) was obtained in the discussion of effective medium theory. It must be clearly explained in the manuscript.
